# A Network-Based Approach for Identification of Subtype-Specific Master Regulators in Pancreatic Ductal Adenocarcinoma

**DOI:** 10.3390/genes11020155

**Published:** 2020-02-01

**Authors:** Yuchen Zhang, Lina Zhu, Xin Wang

**Affiliations:** 1Department of Biomedical Sciences, City University of Hong Kong, Hong Kong SAR, China; yuczhang9-c@my.cityu.edu.hk (Y.Z.); linazhu3-c@my.cityu.edu.hk (L.Z.); 2Shenzhen Research Institute, City University of Hong Kong, Shenzhen 518057, China

**Keywords:** miRNA, regulatory network inference, pancreatic ductal adenocarcinoma, subtype

## Abstract

Pancreatic ductal adenocarcinoma (PDAC), the predominant subtype of pancreatic cancer, has been reported with equal mortality and incidence for decades. The lethality of PDAC is largely due to its late presentation, when surgical resection is no longer an option. Similar to other major malignancies, it is now clear that PDAC is not a single disease, posing a great challenge to precise selection of patients for optimized adjuvant therapy. A representative study found that PDAC comprises four distinct molecular subtypes: squamous, pancreatic progenitor, immunogenic, and aberrantly differentiated endocrine exocrine (ADEX). However, little is known about the molecular mechanisms underlying specific PDAC subtypes, hampering the design of novel targeted agents. In this study we performed network inference that integrates miRNA expression and gene expression profiles to dissect the miRNA regulatory mechanism specific to the most aggressive squamous subtype of PDAC. Master regulatory analysis revealed that the particular subtype of PDAC is predominantly influenced by miR-29c and miR-192. Further integrative analysis found miR-29c target genes LOXL2, ADAM12 and SERPINH1, which all showed strong association with prognosis. Furthermore, we have preliminarily revealed that the PDAC cell lines with high expression of these miRNA target genes showed significantly lower sensitivities to multiple anti-tumor drugs. Together, our integrative analysis elucidated the squamous subtype-specific regulatory mechanism, and identified master regulatory miRNAs and their downstream genes, which are potential prognostic and predictive biomarkers.

## 1. Introduction

Pancreatic ductal adenocarcinoma (PDAC), the major type of pancreatic cancer, has been reported with the worst prognosis among all cancers [1], with five-year survival less than 8%. The lethality of this disease is mainly due to its late presentation, preventing diagnosis at an early stage [2]. Worldwide, the incidence of PDAC is 10 per 100,000 individuals, and the mortality of this disease has essentially equaled its incidence for decades [3]. In the last decades, PDAC studies have identified many somatic mutations [4,5], signaling pathways [6,7,8], along with microenvironmental factors [9,10,11], laying the genetical, biological and ecological foundation of this disease. From the perspective of precision medicine [12,13], all major malignancies are heterogeneous diseases; therefore, dissecting the intrinsic biological heterogeneity in relation to clinical outcomes is of utmost importance for selection of patients for optimized therapy and design of novel therapeutic agents. 

For PDAC the identification of molecular subgroups lags behind, mainly because of the difficulties in the collection of tumor samples that contain an adequate amount of tumor cells rather than the large amount of dense stroma, which is considered as a hallmark of PDAC [14]. Nevertheless, in recent years molecular subtyping studies for PDAC are accumulating [15,16,17]. One of the most representative subtyping system was proposed by Bailey et al. in 2016 [17], by which 4 subtypes that correlated with histopathological characteristics were identified based on the gene expression data of 96 samples: squamous, pancreatic progenitor, immunogenic, and aberrantly differentiated endocrine exocrine (ADEX). Importantly, the squamous subtype of PDAC associates with worse prognosis signifying the importance to focus on these patients. However, despite the recent research progress and insights into PDAC heterogeneity, the regulatory mechanism underlying the squamous PDAC subtype remains largely elusive.

MicroRNAs (miRNAs) are short (about 22 nucleotides) noncoding RNAs that regulate their target genes by either silencing or degrading them through the complementary seed sequences of 3’ strand in the post-transcriptional level [18]. As a result, the expression levels between a miRNA and its target gene often show an inverse correlation [19]. Individual miRNA can target and regulate a group of genes simultaneously [20,21], showing their ability to influence expression of regulatory networks defining a particular cancer subtype. In fact, miRNAs have been implicated for their functions to drive poor prognosis cancer subtypes by regulating the expression of genes actively involved in epithelial–mesenchymal transition (EMT), a process in which loss of epithelial hallmarks and acquisition of mesenchymal features can be observed in cancer cells, enabling them to metastasize to distant organs [22,23]. Furthermore, it is well accepted that transforming growth factor-β (TGFβ) is a robust EMT inducer both under physiological and pathological circumstances [24]. As it has been reported that the squamous subtype of PDAC is associated with TGFβ activation [17], we hereby focus on the key miRNAs as well as their targets to dissect their regulatory mechanism specifically in the squamous PDAC subtype based on *in silico* analysis of multi-omics data.

Using multi-dimensional network inference based on integration of gene and miRNA expression profiles, we have successfully identified subtype-specific master regulatory miRNAs in colorectal cancer [25] and ovarian cancer [26] in our previous studies. In this study, we employed the established network-based approach to gain insights into the molecular determinants of the squamous subtype of PDAC (Figure 1). The regulatory network identified in the squamous subgroup is predominantly regulated by miR-29c and miR-192. Through integrative analysis with the predicted miRNA targets in public databases, we identified potential direct target genes (LOXL2, ADAM12 and SERPINH1) of these two miRNAs. Furthermore, we demonstrated the clinical relevance of these miRNA target genes based on survival analysis in two independent datasets and drug sensitivity analysis in cell lines. Our work suggests that the network-based approach is an efficient strategy to dissect the regulatory mechanism and prioritize potential therapeutic targets specifically for the squamous subtype of PDAC. 

## 2. Materials and Methods 

### 2.1. Data Collection

Public datasets analyzed in this study, including mRNA expression, miRNA expression, 450K DNA methylation microarrays and corresponding clinical information, were downloaded from The Cancer Genome Atlas (TCGA, https://www.cancer.gov/about-nci/organization/ccg/research/ structural-genomics/tcga) by R package ‘TCGAbiolinks’ [27]. Altogether, 149 samples in the TCGA cohort have miRNA, and gene expression data as well as subtyping assignment by Bailey et al. [17]. For validation, gene expression and clinical information of dataset ‘Bailey’ consisting of 70 samples with high tumor purity were downloaded from the supplementary of Bailey et al [17]. Another independent public gene expression dataset (Netherlands cohort, *n* = 90) was download from EMBL-EBI ArrayExpress (E-MTAB-6830). Furthermore, 21 PDAC cell lines with both gene expression and drug sensitivity profiles were downloaded from Cancer Cell Line Encyclopedia (CCLE) (https://portals. broadinstitute.org/ccle) for drug response analysis.

### 2.2. Differential Gene and miRNA Expression Analysis

The RNA-seq and miRNA-seq data downloaded from TCGA was annotated and log-transformed. Genes with duplicated records were filtered by keeping the one with the maximum average expression. Differential gene and miRNA expression analysis was performed using R package ’limma’ [28,29] between squamous subtype and non-squamous subtypes (Appendix A).

### 2.3. Gene Set Enrichment Analysis

To obtain a landscape of biological processes associated with PDAC subtypes, gene set enrichment analysis was performed by R package ‘HTSanalyzeR2’ with permutation of 100,000 times [30]. More specifically, we focused on gene sets including canonical signatures and pathways, metabolic pathways, immune signatures [31] as well as immune and stromal contents calculated by ESTIMATE [32].

### 2.4. miRNA-mRNA Regulatory Networks Inference

Regulatory network inference was performed to study the relationship between miRNAs and potential target genes by integrative analysis of gene and miRNA expression profiles using the ‘RTN’ package [33,34]. More specifically, the network analysis involves three steps: (i) compute mutual information (MI) between a miRNA and all potential targets, removing non-significant associations by permutation analysis; (ii) remove unstable interactions by bootstrapping; and (iii) apply the ARACNe algorithm [35] to reduce redundant indirect regulations. The ARACNe algorithm employs the data processing inequality (DPI) theorem to enrich the regulons with direct interactions between regulator and targets, creating a DPI-filtered transcriptional regulatory network (TRN). Briefly, the DPI theorem assumes that given three random variables, X, Y and Z that form a network triplet interaction network X->Y->Z, and no alternative path exists between X and Z, the information transferred between Y and Z is always larger than the information transferred between X and Z. Based on this assumption, the ARACNe algorithm scans all triplets formed by two microRNAs and one target gene and removes the edge with the smallest MI value of each triplet, which is regarded as a redundant association.

To this end, the expression profiles of differentially expressed miRNAs and genes were combined to perform regulatory network inference. Statistical significance was determined by Benjamini-Hochberg-adjusted (or BH-adjusted) *p* < 0.05 and log_2_ fold change (or log_2_ FC) < −0.5 for calling differential miRNAs and log_2_ FC > 0.25 for calling differential genes. These thresholds were selected mainly under the consideration of the negative correlation between the miRNAs and target genes.

### 2.5. Master Regulator Analysis

Master regulator analysis (MRA) was conducted by a hypergeometric test for overrepresentation of TGFβ signature genes [31] in the regulon of each miRNA. After the hypergeometric tests result for all miRNAs, adjusted *p*-values were calculated using Benjamini-Hochberg procedure. MiRNAs whose regulons contain more than 15 genes were retained for further analysis, since if the regulon size is too small the biological influence of the miRNA would be difficult to interpret statistically.

### 2.6. Prioritization of miRNA Target Genes

Identification of the direct target genes of the prioritized master regulators was based on a strategy looking for the intersection between four genes sets: TGFβ signature genes, predicted functional target genes of the miRNAs by network inference and predicted physical target genes of the miRNAs by mirDIP [36] and miRDB [37] databases, respectively. Spearman correlation analysis was employed to evaluate the correlation between miRNAs expression and their target genes expression.

### 2.7. Survival and Drug Sensitivity Analysis

To inspect the clinical relevance of master regulatory miRNA target genes, Kaplan-Meier analysis was performed to show survival difference between patient subgroups stratified based on the gene expression level. Quantile was used to divide patients into two groups (>25% quantile, High; <25% quantile, Low). The statistical significance was assessed by a log-rank test. Univariate Cox regression analyses were performed to calculate hazard ratios (HRs) with 95% confidence intervals (CIs) to evaluate the prognostic significance as well.

Drug sensitivity analysis was conducted for the 6 targets genes of miR-29c and miR-192 based on 26 PDAC cell lines with both gene expression data and drug response data downloaded from CCLE [38]. For each compound, we separated these 26 cell lines into two subgroups based on the median gene expression of each target gene. Two-sided *t*-tests were used to evaluate the difference in drug sensitivity (measured by IC50) between the two subgroups. Area under the curve (AUC) was calculated by using the expression level of a gene to predict whether a cell line is drug sensitive or resistant, which was based on stratification of IC50 values of all cell lines at the mean.

## 3. Results

### 3.1. Multi-Dimensional Regulatory Network Inference Based on Integrative Analysis of Gene and miRNA Expression Profiles

Using both the TCGA and Bailey gene expression datasets, we performed gene set enrichment analysis (GSEA) to obtain a landscape of biological processes associated with each PDAC subtype (Appendix A). As a result, we found that the four molecular subtypes indeed showed distinct molecular characteristics as defined by Bailey et al [17]. Importantly, the subtype-specific molecular properties are consistent in both the TCGA and Bailey datasets (Figure 2). The squamous subtype was reported to be the most aggressive one and associated with poor clinical outcome; therefore, we focused on this particular subtype hereafter. We found indeed the squamous subtype is characterized by elevated epithelial to mesenchymal transition (EMT), VEGF and TGFβ pathways (Figure 2). In particular, the TGFβ signature genes are almost all significantly upregulated in squamous subtype in TCGA dataset, compared to non-squamous subtypes (Figure 2f,g). 

Focusing on the squamous subtype of PDAC, we employed a network-based approach to interrogate the subtype-specific regulatory mechanism. Based on the TCGA dataset, we first performed differential gene and miRNA expression analysis, comparing the 31 squamous tumors with the other 118 non-squamous tumors. As a result, 44 miRNAs significantly downregulated in the squamous subtype (BH-adjusted *p* < 0.05, log_2_ FC < −0.5) were prioritized as potential regulators, and 1486 genes significantly upregulated in the squamous subtype (BH-adjusted *p* < 0.05, log_2_ FC > 0.25) were considered as potential target genes. Based on expression profiles for these prioritized miRNAs and genes, we inferred a miRNA regulatory network using ‘RTN’ package [33,34]. The network was reconstructed by computing the mutual information (MI) between a miRNA and all potential targets, followed by filtering out unstable interactions by bootstrapping and applying the ‘ARACNe’ algorithm [35] (Figure 3). 

### 3.2. Identification of Putative Master Regulatory miRNAs in the Squamous Subtype of PDAC

Although TGFβ has been implicated in PDAC, the subtype-specific regulatory mechanism remains elusive. To identify potential key regulators specifically for the squamous subtype, master regulator analysis was subsequently performed based on hypergeometric tests (Table 1). Interestingly, we found 19 and 11 TGFβ signature genes enriched in the regulons of miR-29c (BH-adjusted *p* = 6.3 × 10^−06^) and miR-192 (BH-adjusted *p* = 3.0 × 10^−03^), respectively (Figure 4a). Indeed, compared to non-squamous subtypes, we observed significantly lower expression of these two candidate miRNAs (*p* = 1.86 × 10^−05^ for miR-29c and *p* = 1.2 × 10^−04^ for miR-192, Wilcoxon signed-rank tests) in the squamous subtype in TCGA dataset (Figure 4b,c). Importantly, gene ontology analysis for both the two miRNAs’ regulons identified significant biological processes and signaling pathways related to cancer development and metastasis such as: TGFβ pathway, extracellular matrix organization, focal adhesion, TGFβ, angiogenesis and cell migration (Appendix A). Furthermore, the functional relevance of the two miRNAs in TGFβ signaling pathway regulation has been reported in the literature. More specifically, miR-192 is a critical downstream mediator of TGFβ/Smad3 signaling in the development of renal fibrosis [39]. TGFβ can inhibit miR-29c expression, leading to Wnt activation in pancreatic cancer [40]. However, our results for the first time showed that the two miRNAs are master regulators of TGFβ signaling pathway specifically in the squamous subtype of PDAC.

### 3.3. Prioritization of Potential Therapeutic Targets by Identification of the Direct Target Genes of the Master Regulatory miRNAs 

As miRNAs are hard to target for design of novel therapeutic interventions, we next sought to explore their direct downstream target genes that are potentially more druggable. Through the intersection strategy, five genes were identified as the direct targets of miR-29c, and one target gene of miR-192, respectively (Figure 5a,b). All these target genes have been reported for their function or association with PDAC in the literature (detailed review in Table 2). For instance, LOXL2 was recently identified as the direct downstream target of specific protein 1 (SP1) regulating EMT in PDAC [41].

Using Spearman correlation analysis, we found that in the TCGA dataset the expression levels of miR-29c and miR-192 were significantly inversely correlated their corresponding target genes (Appendix A; Figure 5c,d). Among all the relationships, the expression of LOXL2 showed the highest inverse correlation with the expression of miR-29c (Spearman correlation coefficient, or Rho = −0.527, *p* < 1.0 × 10^−16^, Figure 5c). The expression levels of miR-192 and SERPINE1 are also significant inversely correlated (Rho = −0.406, *p* = 7.02 × 10^−07^, Figure 5d). Together, we identified six direct target genes that are functionally inversely correlated with miR-29c and miR-192, providing multiple potential novel therapeutic targets for the squamous subtype of PDAC.

### 3.4. Exploring the Upstream Regulatory Mechanism of the Master Regulatory miRNAs

To possibly explain the significant lower expression of miR-29c and miR-192 in the squamous subtype of PDAC, we also attempted to look for potential upstream mechanisms. We speculated that hypermethylation of the promoters maybe potentially responsible for suppression of these two miRNAs. By integrative analysis of 450K methylation microarray data and miRNA expression profiles in the TCGA dataset, we did not find significant difference in promoter methylation between squamous and non-squamous subtypes (Appendix A). Furthermore, significant negative correlation was observed between promoter methylation and expression of miR-192 (Appendix A), but not miR-29c (Appendix A). Indeed, low expression of miR-192 in PDAC was shown to be caused by hypermethylated promoters [47]. Consistent with our finding, SERPINE1, a target gene of miR-192, was also validated in this work. Together, these results showed that elevated DNA methylation could partially explain the significant low expression of the prioritized master regulatory miRNAs. 

### 3.5. Clinical Relevance of the Master Regulatory miRNA Target Genes 

Survival analysis was performed to investigate the potential clinical relevance of master regulatory miRNA target genes. Based on univariate Cox regression analysis, we found that indeed the expression of LOXL2 (HR = 1.42, *p* = 1.69 × 10^−02^), ADAM12 (HR = 1.37, *p* = 4.80 × 10^−02^), LIMS1 (HR = 1.65, *p* = 2.71 × 10^−03^) and SERPINH1 (HR = 1.50, *p* = 1.28 × 10^−02^) all showed a significant association with overall survival (Table 3), respectively. To further examine the robustness of the prognostic power of these genes, an independent cohort (termed the Netherlands dataset hereafter) was analyzed for validation. Consistent with the Bailey dataset, we found that indeed in the Netherlands dataset LOXL2 (HR = 1.31, *p* = 3.24 × 10^−03^), ADAM12 (HR = 1.25, *p* = 1.02 × 10^−03^) and SERPINH1 (HR = 1.59, *p* = 2.00 × 10^−03^) also showed significant prognostic value (Table 4). Furthermore, Kaplan–Meier curves also demonstrated that PDAC patients with higher expression of master regulatory miRNA target genes showed poorer overall survival in both the Bailey and Netherlands datasets (Figure 6; Appendix A). *p* values derived from Log-rank tests are all significant except LIMS1 in the Netherlands dataset.

Importantly, both LOXL2 and SERPINH1 have already been implicated as therapeutic targets in other diseases. For instance, it has been reported that selective targeting of LOXL2 could suppress hepatic fibrosis progression and contribute to its reversal [48], and more excitingly, the clinical trial has been announced online [49]. Although tested for treatment of hepatic fibrosis, if the drug turns out to be successful in future, it may provide a new therapy for PDAC patients. The inhibitor for SERPINH1 has been noted in mature human drug targets [50] and has been commercialized for treatment of osteogenesis imperfecta. Together, our analyses demonstrated the potential to exploit the master regulatory miRNA target genes as therapeutic targets of PDAC and set the stage for further mechanistic study and experimental validations.

### 3.6. Drug Sensitivity Analysis 

To further investigate the potential to use the master regulatory miRNA target genes as predictive biomarkers, we analyzed drug sensitivity data for PDAC cell lines obtained from CCLE [38]. The total 26 cell lines were stratified into two subgroups based on the median gene expression of each target gene. More specifically, for each compound (drug) a two-sided *t*-test was employed to measure the statistical significance of differential drug response based on IC50, which is a quantitative indicator of the drug effect (Appendix A). Interestingly, we found cell lines with high expression of these miRNA target genes showed significantly lower sensitivities to multiple anti-tumor drugs (Figure 7) including AEW541, which is a IGF-1R inhibitor [51], AZD0530 (or Saracatinib), which is a dual kinase inhibitor (Src inhibitor and Bcr-Abl tyrosine-kinase inhibitor) [52], ZD-6474 (or Vandetanib), which is a kinase inhibitor of a number of cell receptors, mainly VEGFR, EGFR and the RET-tyrosine kinase [53], AZD6244 (or Selumetinib), which is a MEK inhibitor, as well as chemotherapy of Topotecan [54]. Furthermore, all these miRNA target genes also showed a potential to be predictive biomarkers (all AUC ≥ 0.73, Appendix A). Especially, LOXL2 and SERPINH1 demonstrated high performance for predicting the sensitivity of ZD-6474 (AUC = 0.83 and 0.87, Appendix A). 

## 4. Discussion

Recent molecular subtyping studies about PDAC have gained novel insights into the inter-tumor heterogeneity of this deadly disease [15,16,17], paving the way for better understanding the mechanism and facilitating stratification of patients for better clinical management. Although the consensus between the multiple subtyping systems is yet to be reached, the common conclusion is that PDAC should no longer be treated as a single disease, but multiple subtypes associated with distinct biological and clinical characteristics. However, despite the community effort on PDAC classification, subtype-specific molecular mechanism remains to be explored. Focusing on the squamous subtype of PDAC defined by Bailey et al. [17], which is associated with poor prognosis, we sought to elucidate the underlying regulatory mechanism in this study. 

Based on integrative analysis of multi-omics data in the TCGA dataset, we inferred a miRNA regulatory network followed by identification of master regulatory miRNAs that are key to activation of EMT. The multi-dimensional network inference approach based on ARACNE was established earlier and has been successfully applied to identify master regulatory miRNAs in colorectal cancer and ovarian cancer in our previous studies [25,26]. To investigate the stability of ARACNE, for demonstration we fixed the cutoff for miRNA selection (−0.5) and varied the cutoff on log_2_ FC (0.25, 0.5, 0.75, 1) for gene selection. We found that a higher cutoff for gene selection resulted in a smaller network, which would further make it difficult to do master regulator analysis based on hypergeometric tests (not enough sample size for the tests). Therefore, we decided to include as many genes as possible in the first place by setting a relatively low threshold on log_2_ FC (0.25). However, no matter which cutoff we chose, we consistently identified the same master regulator miRNA(s) (Appendix A), which suggests the reliability of ARACNE in identifying the master regulators.

Using the network analysis, we identified two master regulatory miRNAs, miR-29c and miR-192, regulating TGFβ signaling pathway, which is potentially responsible for the promoted EMT in the squamous subtype of PDAC. Meanwhile, through the integrative analysis of DNA methylation and miRNA expression profiles, we could preliminarily conclude that elevated promoter methylation only partially explains the low expression of miR-29c and miR-192 in the squamous subtype of PDAC. Genetic and epigenetic regulations other than DNA methylation may exist, and further investigation is needed to further elucidate the upstream regulatory mechanisms of these two master regulatory miRNAs. On the other hand, since miRNAs are difficult to target for developing therapies, we further identified the direct downstream target genes of these two miRNAs, which showed prognostic value in multiple public datasets. Drug sensitivity analysis further demonstrated the potential for the miRNA target genes to be employed as predictive biomarkers for optimized therapeutics for PDAC patients. Furthermore, inhibitors have been developed for targeting LOXL2 and SERPINH1 in other diseases, demonstrating the potential to consider these miRNA target genes as drug targets for PDAC, which warrants further experimental validations.

Since TGFβ pathway involves multiple regulatory mechanisms, other TGFβ associated genes may also be prognostic and/or predictive biomarkers. For comparison, we performed survival analysis for 8 representative signature genes in the TGFβ pathway including SMAD2, SMAD3, SMAD4, TGFB1, TGFB2, TGFB3, TGFBR1 and TGFBR2 based on the Bailey and Netherlands cohorts. Using univariate Cox regression analysis, we found that only TGFB1 and TGFB2 genes show consistent association with overall survival (Appendix A). Although three genes (TGFB1, SMAD3, TGFB3) showed statistical significance in the Kaplan-Meier survival analysis, none of them show consistent significance in both the Bailey and Netherlands cohorts. Furthermore, we found SMAD2, TGFB1, TGFB2, and TGFBR2 showed significant differential sensitivities in several drugs (Appendix A). However, the expression levels of these genes show predictive values on TAE684, THK258, and Erlotinib, which are not the same as we found in the analysis for LOXL2, SERPINH1 and LIMS1. Therefore, other TGFβ associated genes do not represent the same miRNA regulatory mechanism we identified in the study. 

In conclusion, our findings based on multi-dimensional network analysis provide compelling data for better dissection of the molecular mechanism and clinical relevance of specific subtypes of PDAC and set the stage for further experimental validation and further in-depth mechanistic studies in future. 

## Figures and Tables

**Figure 1 genes-11-00155-f001:**
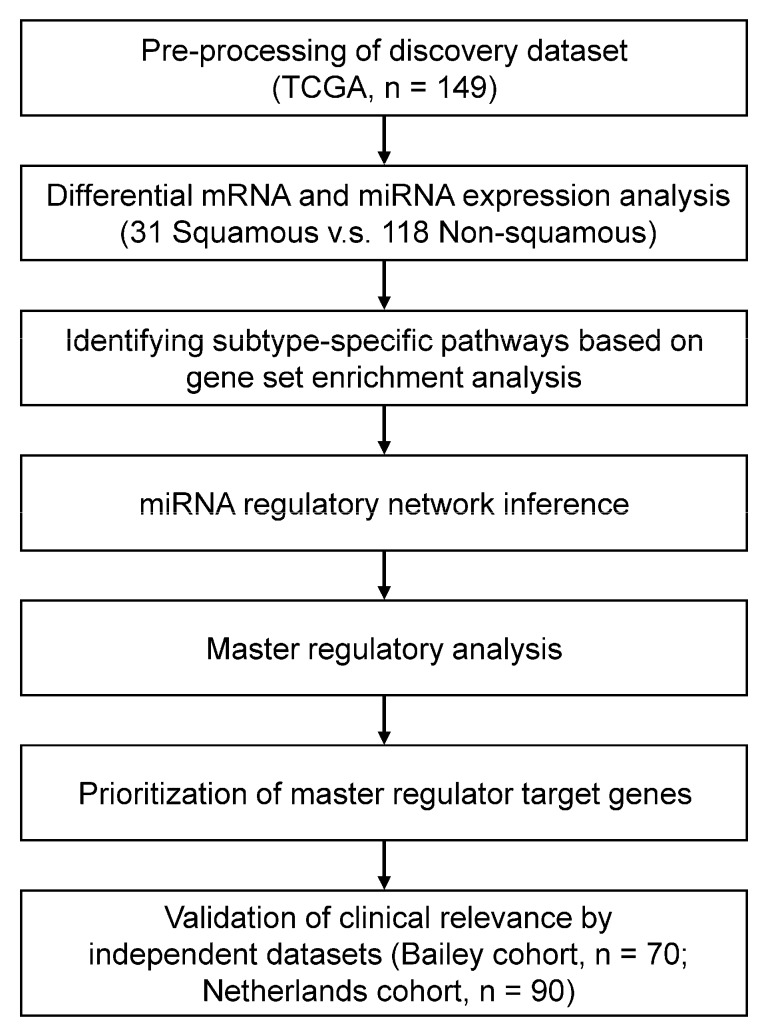
A schematic workflow for dissecting the squamous subtype-specific miRNA regulatory mechanism in PDAC using an integrative network-based approach.

**Figure 2 genes-11-00155-f002:**
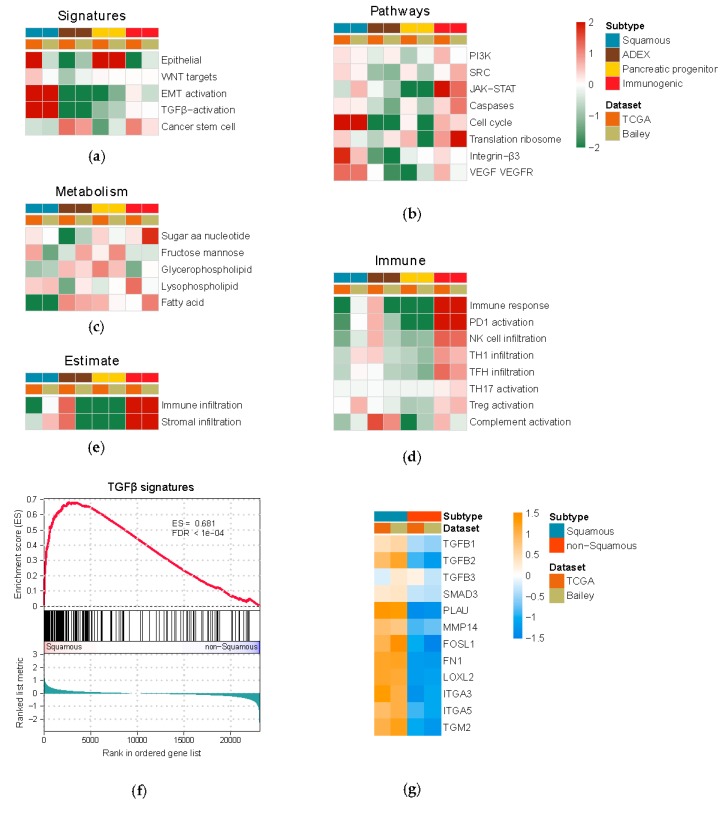
A landscape of molecular properties associated with PDAC subtypes. (**a**–**e**) Heatmaps illustrating gene set enrichment analysis on canonical signatures and pathways, metabolic pathways, immune signatures as well as immune and stromal contents calculated by ESTIMATE [33] (*n* = 149, TCGA dataset; *n* = 70, Bailey dataset). Red color represents over enrichment and green represents under enrichment; the depth of color is proportionate to −log_10_ (*p*-value) of GSEA. (**f**) Gene set enrichment analysis (GSEA) showed TGFβ signature genes are significantly up-regulated in the squamous subtype in TCGA dataset. (**g**) Heatmap showing the average log_2_ fold change of key TGFβ signature genes between squamous and non-squamous subtype in both TCGA and Bailey data set.

**Figure 3 genes-11-00155-f003:**
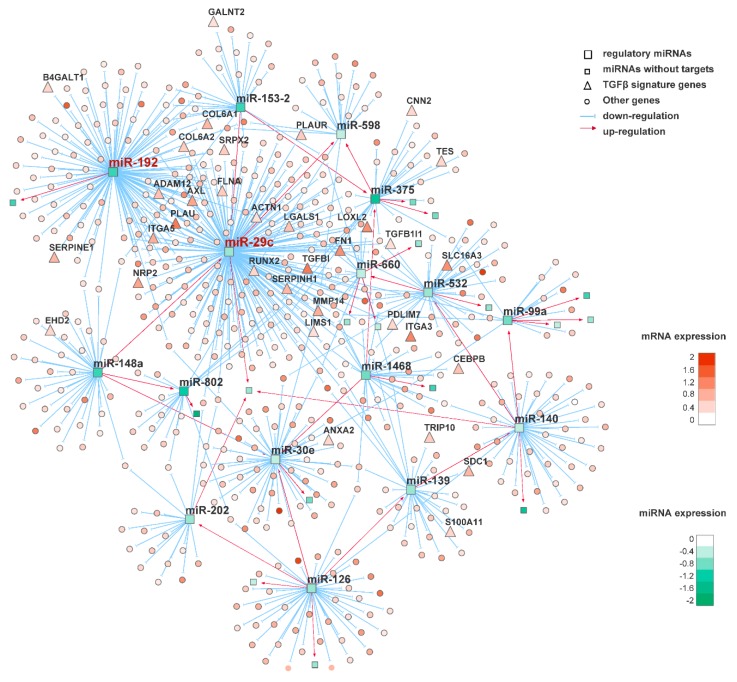
A miRNA regulatory network inferred from integrative analysis of gene and miRNA expression data. miR-29c and miR-192 were identified as the most statistically significant master regulators. Predicted genes regulated by the two miRNAs are colored according to their differential expression levels between squamous and non-squamous samples (green: lowly expressed, red: highly expressed in the squamous subtype). TGFβ signature genes are denoted as triangles.

**Figure 4 genes-11-00155-f004:**
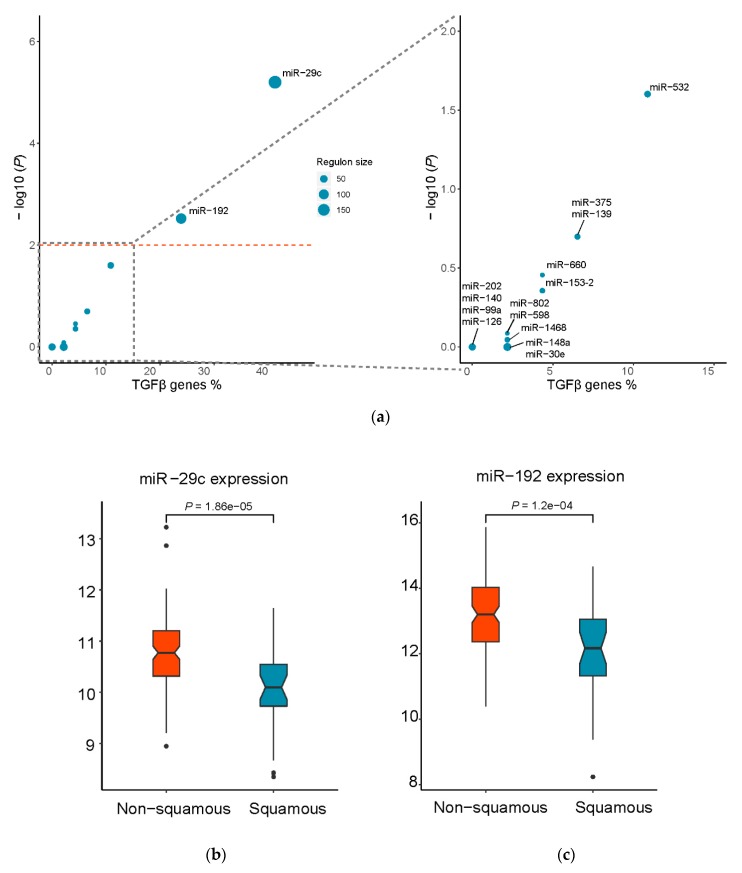
Master regulator analysis identified miRNAs key to the regulation of the TGFβ signaling pathway. (**a**) Prioritization of putative master regulators based on the statistical significance of overrepresentation of the regulon of each miRNA for TGFβ signature genes (−log_10_ transformed BH-adjusted *p*-value, hypergeometric tests) and the proportion of TGFβ genes regulated by a miRNA; **(b,c**) Compared to non-squamous subtypes, miR-29c and miR-192 are significantly downregulated in the squamous subtype in TCGA dataset (*n* = 31 for squamous, *n* = 118 for non-squamous, the *p*-values were corrected using the Benjamini-Hochberg method).

**Figure 5 genes-11-00155-f005:**
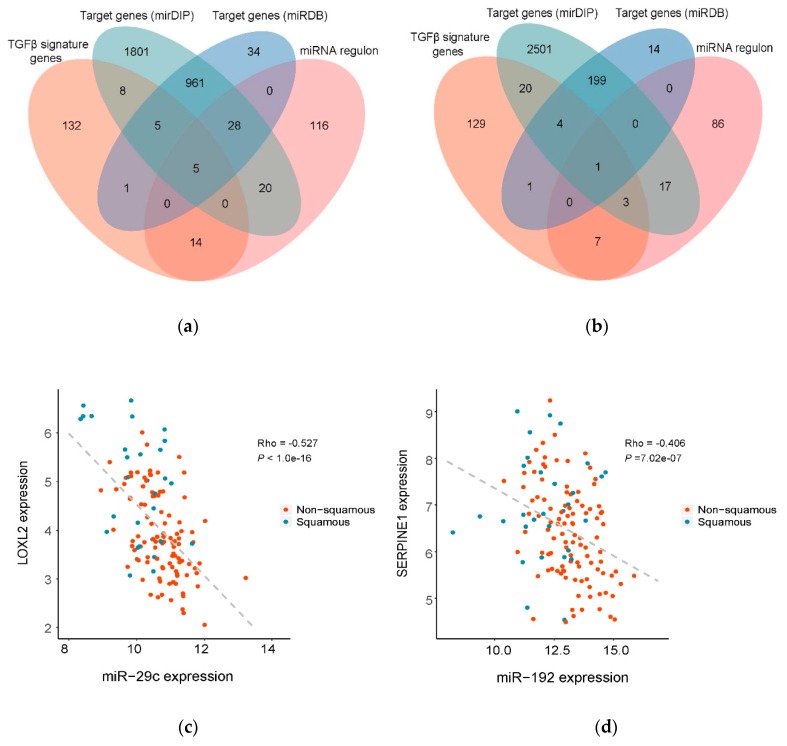
Identification of downstream target genes of miR-29c and miR-192 mediating TGFβ signaling pathway. (**a,b**) Venn diagram illustrating target gene prediction for miR-29c and miR-192 respectively by taking the intersection between predicted physical target genes by miRDB and mirDIP databases, TGFβ signature genes and predicted functional target genes in the regulon of miRNAs in the regulatory network. (**c,d**) Significant inverse correlation between two candidate target genes, LOXL2 and SERPINE1, and corresponding master regulatory miRNAs. The expression profiles were from the TCGA dataset. Rho: Spearman correlation coefficient.

**Figure 6 genes-11-00155-f006:**
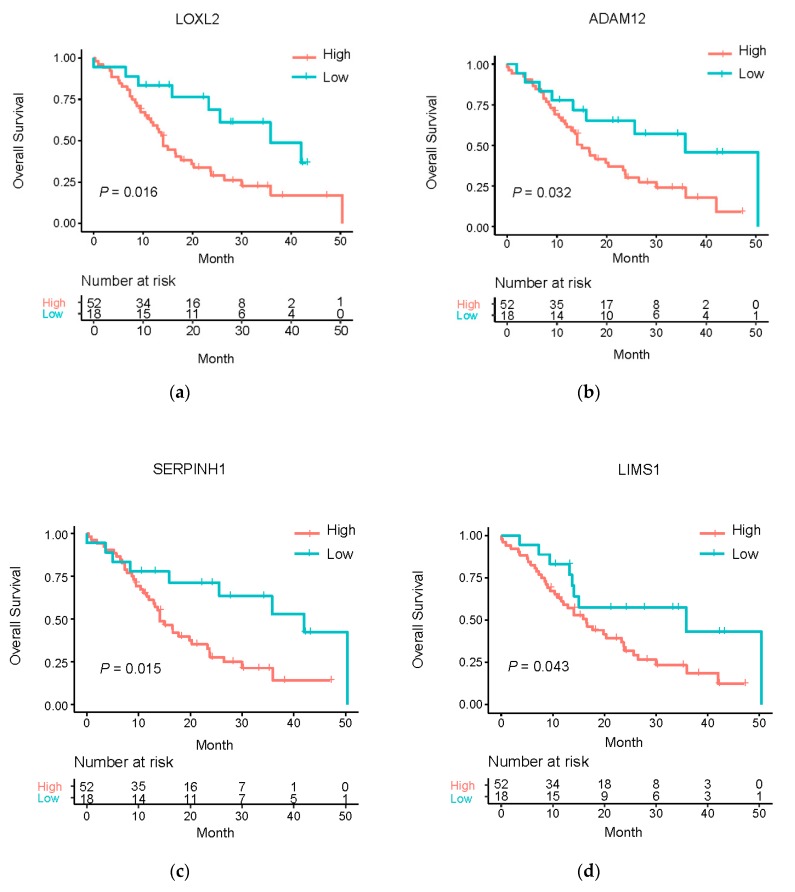
Kaplan-Meier survival analysis showing the clinical relevance of the target genes regulated by miR-29c in the Bailey dataset. (**a,b**) Kaplan–Meier curves illustrating the overall survival of PDAC patients in the two subgroups stratified based on the expression of LOXL2 and ADAM12, respectively (cutoff: 25% percentile). (**c,d**) Kaplan–Meier curves illustrating the overall survival of PDAC patients in the two subgroups stratified based on the expression of SERPINH1 and LIMS1, respectively (cutoff: 25% percentile).

**Figure 7 genes-11-00155-f007:**
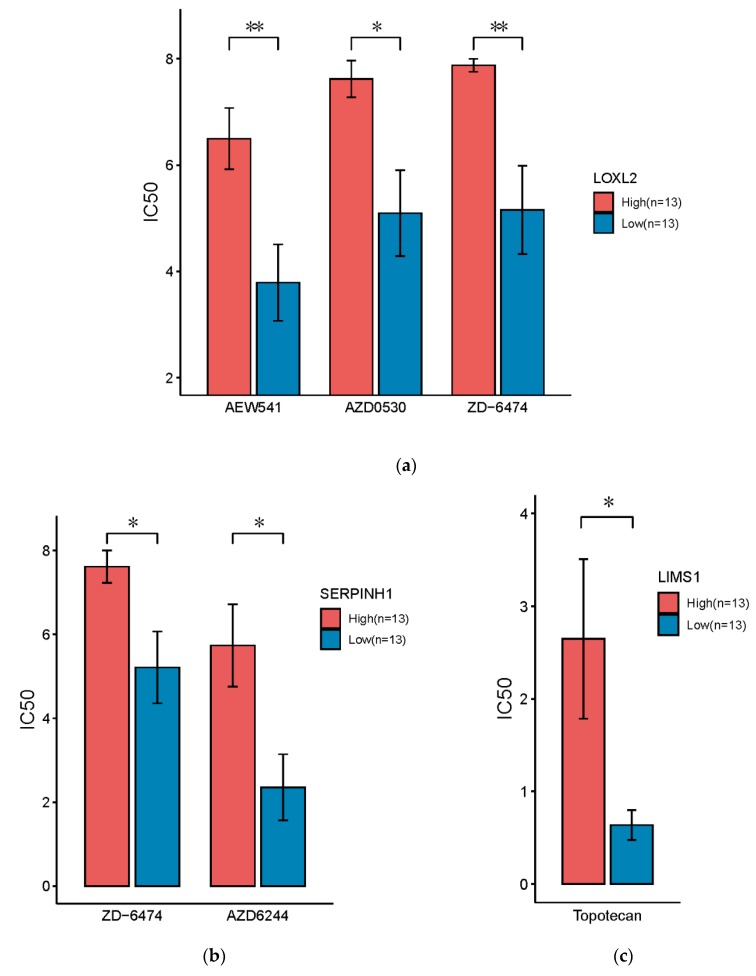
Drug sensitivity analysis based on PDAC cell lines showed the expression of miR-29c target genes were associated with the effect of anti-tumor compounds (drugs). Drug sensitivity was quantified based on the IC50 index. Lower expression of LOXL2 (**a**), SERPINH1 (**b**), and LIMS1 (**c**) were significantly more sensitive to anti-tumor drugs (*p*-values are based on two-tailed *t*-tests, * *p* < 0.05, ** *p* < 0.01).

**Table 1 genes-11-00155-t001:** Master regulator analysis of miRNAs in the squamous subtype of PDAC (Only showing the miRNAs with the adjusted *p*-value < 0.01).

Regulon	Universe Size	Regulon Size	Total Hits	Expected Hits	Observed Hits	*p*-value	Adjusted *p*-value
miR-29c	1530	187	46	5.62	19	4.00 × 10^−07^	6.30 × 10^−06^
miR-192	1530	115	46	3.46	11	3.80 × 10^−04^	3.00 × 10^−03^

**Table 2 genes-11-00155-t002:** Target genes directly regulated by master regulatory miRNAs with reported functions in PDAC.

Regulator	Target Genes	Log_2_ FC (Squamous v.s. Non-Squamous)	Adjusted *p*-Value	Reported Function	References
**miR-29c**	COL6A2	0.58	2.43 × 10^−02^	One SNP in COL6A2 (21q22.3) is associated with a higher risk of pancreatic cancer	[42]
	SERPINH1	0.70	2.37 × 10^−05^	Interacts with HMP19, implicated in PDAC invasion and progression	[43]
	LOXL2	1.19	3.91× 10^−06^	Specific protein 1 (SP1) regulates EMT via LOXL2 in PDAC	[41]
	ADAM12	0.66	1.53× 10^−02^	A circulating marker for stromal activation in PDAC and can predict response to chemotherapy	[44]
	LIMS1	0.33	7.74× 10^−03^	Promotes pancreatic cancer cell survival under specific conditions	[45]
**miR-192**	SERPINE1	0.78	1.17× 10^−02^	Mediates tumor invasion and metastasis and acts as a prognostic marker	[46]

**Table 3 genes-11-00155-t003:** Univariate Cox regression analysis of miR-29c target genes based on the Bailey cohort.

Gene Symbol	Beta	HR (95% CI for HR)	Wald Test	*p*-Value
LOXL2	0.35	1.42 (1.06–1.89)	5.71	1.69 × 10^−02^
ADAM12	0.31	1.37 (1.00–1.87)	3.91	4.80 × 10^−02^
COL6A2	0.18	1.19 (0.84–1.70)	1.48	3.25 × 10^−01^
SERPINH1	0.41	1.50 (1.09–2.07)	6.20	1.28 × 10^−02^
LIMS1	0.50	1.65 (1.19–2.29)	9.00	2.71 × 10^−03^

HR: hazard ratio; CI: confidence interval.

**Table 4 genes-11-00155-t004:** Univariate Cox regression analysis of miR-29c target genes based on the Netherlands cohort.

Gene Symbol	Beta	HR (95% CI for HR)	Wald Test	*p*-Value
LOXL2	0.27	1.31 (1.10–1.58)	8.67	3.24 × 10^−03^
ADAM12	0.22	1.25 (1.09–1.43)	10.80	1.02 × 10^−03^
COL6A2	0.37	1.45 (1.14–1.84)	8.98	2.73 × 10^−03^
SERPINH1	0.47	1.59 (1.19–2.14)	9.55	2.00 × 10^−03^
LIMS1	0.33	1.39 (1.01–1.93)	4.01	4.53 × 10^−02^

HR: hazard ratio; CI: confidence interval.

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
