# Peer review of "A Network-Based Approach for Identification of Subtype-Specific Master Regulators in Pancreatic Ductal Adenocarcinoma"

_genes, 2020, doi:10.3390/genes11020155_

Round 1

Reviewer 1 Report

The authors of "A network-based approach for identification of master regulatory microRNAs and prioritization of novel therapeutic targets specifically for the squamous subtype of pancreatic ductal adenocarcinoma"  analyzed three public available datasets using integrative approaches. They performed several analysis, including replication, to pinpoint to potential master regulatory miRNAs and their target genes. I find the paper interesting, though revisions are needed to present the workflow and results clearer.

In general, it is concluded that the regulatory mechanism is elucidated, though the mechanism suggested is already presented in the introduction as shown by others. The authors do find miRNAs and their targets, which raises the question whether based on those novel results, the biological mechanisms involved can be further revealed. The manuscript is long (including title and introduction), and shortening of it and e.g. limiting the number of (supplementary) figures may benefit the manuscript.

Comments per section:

Introduction
Long introduction that could be shortened.

Materials and Methods
Line 105-107: 149 samples of which only 70 were reliable...: in the results all 149 were used. Why?

Line 121-123: Thresholds for differential expression: why are there different log2FC thresholds? Why not both 0.25 or 0.50? Further, negative correlation should be explained better (in M&M or in introduction)

In the results section, the GSEA is described before. It would help if the order in the M&M is matching.

Line 132-146: How reliable is the network construction with ARACNE and what is the effect of changing thresholds? Can network results be replicated e.g. with another method?

Line 150-151: Should be placed in the results

Results are blended in the methods-section, e.g. Line 153 (miR-29c and miR-192).

Line 155: what overlap? presence of it in the different sets?

Results

General: a lot of repeat from the methods section.

Line 174-186: It is unclear from the setup of the manuscript why the GSEA is performed, as results only confirm what is described in the introduction. The introduction sets up for a manuscript only looking into the squamous subtype, but this first part is looking into all subtypes.

Line 189-191: what are results without FC threshold?

MRA/Table 1: It is unclear what analyses is performed, i.e. which targets are used? The DE genes, a list of EMT and TGFbeta signature genes, ...? The methods indicates EMT and TGF, the results indicate TGF, but why are the DE genes not used as they are identified as potential targets.

Table 1: only present significant results in main tables. 

Line 214-215: is there any multiple testing correction performed?

Figrue 4A: due to the many lines/miR-names badly readable

Line 246-247: are expression levels normally distributed that Pearson is most correct estimate for the correlations?

Table 2: switch column 1 and 2 would increase readability

Drug sensitivity analysis: Is any multiple testing correction performed?

Discussion

Line 327-330: Focus of the results has been on individual miRNAs and their targets, not on the actual biological mechanism involved. Could the biological mechanism behind it be investigated? I.e. the network gives a cluster of genes that is connected to the miRNA regulators, are they in the same biological pathway that could explain more on the actual mechanism differentiating squamous PDAC? Further, in line 335 a mechanisms is stated: is this purely based on previous studies? Can this study add anything except for the target genes?

From Line 341: methylation data has not been described in the results?

What is the clinical relevance of this study? E.g. when the target genes would be used in clinics, what is the sensitivity etc of testing the expression levels of those genes in patients with PDAC?

Reviewer 2 Report

The authors have put together an interesting manuscript aimed at further elucidating the genomic differences between subsets of pancreatic ductal adenocarcinoma.  The approach of employing a bioinformatic analysis to discover master regulators of significant processes is innovative and productive.  While the concept that increased TGF beta signaling is a driver of a poor outcome cancer subtype by itself is unsurprising, the clinical validation of the miRNA regulation is significant.  There are some points that I suggest the authors address.

1) Line 107-110 are a bit confusing. Please clarify the number of samples used in the discovery stage.  Were 149 samples included or were only the 70 samples with relative reliability included?  If all 149 were included in discovery, please add language and rational clarifying this.

2)The Kaplan - Meier survival analysis (clinical relevance) should demonstrate utility of the specific biomarkers against other TGF-b associated genes. The argument proposed seems to imply that through the bioinformatic analysis the authors have identified genes with significant effects on disease progression.  The question I see is are these genes simply representative of the TGF beta pathway overall or are these genes overly impactful?  I suggest running the same survival analysis on genes like TGFb and its targets. 

3) I question the purpose of the drug sensitivity analysis. Are the authors suggesting that their identified genes (LOXL2, SERPINH1, LIMS1) are novel biomarkers for drug resistant tumors or are they suggesting that these proteins drive the observed resistance?  In either case I suggest a comparison to other TGFb genes.  For example, if the suggestion is that tumors can be stratified by LOXL2, then stratify the same cells based on expression of TGFb and its targets.  If this can be done, then perhaps the genes themselves are just representative of the TGFb pathway and thus stratification of tumors should be done by a larger panel of TGFbeta targets rather than LOXL2.

4) Overall in terms of clinical relevance I am not convinced that the downstream targets of the miRNAs would actual be useful as drug targets. The authors argue that miRNAs are not viable targets, but is LOXL2 or SERIPNH1 a good target?  There is no evidence for this as the manuscript stands.  It is not necessary to show this for publication; however, this should be discussed in the discussion if more experiments are not run.

5) There is a secondary methylation analysis that was performed and mentioned in the discussion. I wonder why this was not added to the main manuscript.  As supplemental information it should not be central to our understanding of the findings, however it is a significant portion of the discussion.  I suggest reworking the discussion to not include the methylation analysis or add a figure in the main body of the manuscript for this analysis.

6) There are some grammatical errors in the introduction. This section should be edited by the authors.      

Round 2

Reviewer 1 Report

Thank you for the revisions made on the manuscript. I have a few minor points left:

Line 88-89: The text has been edited as such that all 149 samples were used. However, previously it was described that only 70 were reliable. Could it be elaborated why also unreliable (?) samples were used? Do results change when only reliable samples are used? Table 1: why using a threshold of 0.01 and not 0.05 (as done in the rest of the manuscript)? Figure 4A: In the PDF version I here have, the lines are still crossing the miR-names, making them unreadable Figure 4B/C: I suggest using 'FDR=' in stead of 'P=' in the figure, or to include in the legend they are corrected. Section 3.4: I would suggest rewriting of the 'discussion-paragraph' in such way that in section 3.4 the actual results are presented, and the discussion of results is described in the discussion-section.
